# Characteristics of the Maxillofacial Morphology in Patients with Idiopathic Mandibular Condylar Resorption

**DOI:** 10.3390/jcm11040952

**Published:** 2022-02-11

**Authors:** Kotaro Tanimoto, Tetsuya Awada, Azusa Onishi, Naoki Kubo, Yuki Asakawa, Ryo Kunimatsu, Naoto Hirose

**Affiliations:** 1Department of Orthodontics and Craniofacial Developmental Biology, Graduate School of Biomedical and Health Sciences, Hiroshima University, Hiroshima 734-8553, Japan; azusa130@hiroshima-u.ac.jp (A.O.); neoki39@hiroshima-u.ac.jp (N.K.); yukitann@hiroshima-u.ac.jp (Y.A.); ryoukunimatu@hiroshima-u.ac.jp (R.K.); hirose@hiroshima-u.ac.jp (N.H.); 2Health and Welfare Bureau for the Elderly, Ministry of Health, Labour and Welfare, Tokyo 100-8916, Japan; awada-tetsuya.s66@mhlw.go.jp

**Keywords:** idiopathic mandibular condylar resorption (ICR), temporomandibular joint (TMJ), magnetic resonance imaging (MRI)

## Abstract

Idiopathic mandibular condylar resorption (ICR) is a pathological condition characterized by idiopathic resorption of the mandibular condyle, resulting in a decrease in the size and height of the mandibular condyle. The purpose of this study was to characterize the maxillofacial morphology of ICR patients. Subjects were selected from patients that attended our orthodontic clinic between 1991 and 2019. Twenty-five patients were diagnosed with ICR by magnetic resonance imaging; however, growing patients were excluded. In total, 18 patients were finally selected. The control group comprised 18 healthy volunteers. Lateral and frontal cephalograms were also used. The ICR group had significantly more severe skeletal class II malocclusions than the control group, mainly due to retrusion of the mandible. In the ICR group, there was a tendency for a skeletal open bite due to a significantly larger clockwise rotation of the mandible than in the control group. There was no significant difference between the two groups in the inclination of the upper and lower central incisors or protrusion of the upper and lower central incisors and first molars. ICR patients have been suggested to exhibit skeletal open bite and maxillary protrusion with changes in maxillofacial morphology due to abnormal resorption of the mandibular condyle.

## 1. Introduction

Idiopathic mandibular condylar resorption (ICR) is a pathology of the temporomandibular joint (TMJ) that is characterized by condylar deformation leading to idiopathic and progressive loss of condylar height [1,2]. Among the pathological conditions with mandibular resorption, ICR is diagnosed when the possibility of other diseases is excluded [3]. Pathological conditions related to mandibular condylar resorption including systemic diseases such as collagen disease, dermatomyositis, systemic sclerosis, growth hormone deficiency, and endocrine disorders, as well as local lesions such as trauma and a history of TMJ surgery [4,5,6]. In addition, ICR is clearly distinguished from mandibular condyle dysplasia, in which the growth and development of the mandibular condyle is inhibited for some reason [7]. ICR is localized to the TMJ and is not accompanied by arthritis or synovitis; therefore, it is also clearly distinguished from juvenile arthritis [8].

In the diagnosis of ICR, imaging findings similar to those of TMJ-osteoarthritis (TMJ-OA) are observed, but the resorption is relatively fast, and the progression is generally long term. It has been reported that the progression of ICR may cause occlusion and skeletal muscle instability, leading to maxillofacial deformity and TMJ dysfunction and pain [9,10]. However, TMJ symptoms in ICR are slight in many cases [11], and 25% of cases are asymptomatic [12]. A possible reason for this could be that in ICR mice, enlarged joint space due to hyperplasia of synovial tissue makes the displaced articular disk movement easier during jaw movement [12]. As another study suggested, no abnormalities are found in the synovial tissue of ICR joints, and the details remain unclear [13].

As treatments for ICR, orthognathic surgery, distraction osteogenesis [14], condylectomy and costochondral graft reconstruction [15], and total alloplastic TMJ reconstruction [16,17,18] have been reported. Although effective cases of total alloplastic TMJ reconstruction for ICR joints have been reported [16,17,18], long-term stability and clear indications have not been established. Moreover, an effective method for preventing the onset or suppression of disease progression has not yet been established.

Female hormones [19], orthognathic surgery [20,21], orthodontic treatment, trauma, TMDs, and mechanical stresses on the mandibular condyle due to oral habits [1,2] have been suggested as factors related to ICR pathology. However, the detailed mechanism of ICR onset and progression remains unclear, and it is difficult to prevent ICR and predict prognosis. Furthermore, owing to its low incidence, few statistical studies have been conducted. Alsabban et al. [10] reported that the majority of ICR patients had malocclusion, of which 93.8% were Angle class II malocclusions, and more than 65% had an anterior open bite.

However, the causal relationship between ICR and malocclusion requires further examination. This is because the exact time of onset of the pathological condition is unknown, making it difficult to accurately assess the effect of ICR on maxillofacial growth in most patients. In our department, the onset of ICR has been observed in the early teens; however, the onset of ICR in teens has not been confirmed. Since the number of growing patients in our department is small [22], it is possible that the initial pathological status of ICR has not been completely detected. In addition, since there are individual differences in maxillofacial skeletal growth during the growth period, it is difficult to have an appropriate control group for the young ICR patient group. On the other hand, in adult ICR patients, as the pathologic status progresses, aggravation of skeletal open bite accompanied by mandibular retrusion is often observed, and the pathological condition is relatively easy to detect. The control group was relatively easy to set up. Therefore, in the present study, we examined the characteristics of maxillofacial morphology in ICR mice after the completion of growth.

We hypothesized that the maxillofacial morphology of ICR patients differs from that of individuals with normal occlusion without temporomandibular disorders (TMDs). Since all patients diagnosed with ICR in our department were female [22], patients in their late teens or later whose growth was confirmed to have completed were targeted. This study aimed to compare maxillofacial morphology between ICR patients and control subjects of the same age and sex who had normal occlusion without TMDs.

## 2. Materials and Methods

This study was approved by the Hiroshima University Ethical Review Board for an epidemiologic study (approval number: E-1684).

### 2.1. Subjects

Eighteen female patients diagnosed with ICR who had completed growth and consented to participate in the study were included. Data were extracted from patients (2394 males and 4597 females) who visited the Hiroshima University Hospital Orthodontic Clinic with malocclusion as the chief complaint between 1991 and 2019.

During the first examination, all 6991 subjects underwent clinical examination of the TMJ, including interviews and palpation. TMDs were diagnosed as having one or more symptoms of TMJ noise, pain, or disturbance of jaw movement at the time of TMJ examination and those with a history of TMD. TMJ magnetic resonance imaging (MRI) was performed on 1384 patients (290 men and 1094 women) who were clinically diagnosed with TMDs.

MRI diagnosis of the pathological condition of the TMJ was performed by dentists in the Department of Oral Radiology and Orthodontics.

Of the 309 patients, 268 (3.8% of all orthodontic patients and 86.8% of patients with mandibular condylar deformity) were diagnosed with TMJ-OA, whereas 25 were diagnosed with ICR (0.36% of all orthodontic patients and 8.1% of patients with mandibular condylar deformity). The diagnosis of ICR was made when continuous absorption on both sides or one of the mandibular condyles was shown on TMJ-MRI, and the possibility of other diseases was excluded. Of these, 24 were diagnosed with ICR by TMJ-MRI at the first examination, and growing patients were excluded, with 18 (18 female and 0 male, age, mean ± SD, 25.5 ± 5.3 years) being selected as ICR group subjects in this study.

The control group consisted of 18 female volunteers in fourth grade (ages, mean ± SD, 24.1 ± 1.2 years) from dental students in Hiroshima University School of Dentistry from 2016 to 2019 (total 212 students) who consented to participate in the study and were judged to have normal occlusion and no TMDs. Lateral and frontal cephalograms obtained under the same conditions were used for both the ICR and control groups.

Inclusion criteria: Those who met the above conditions and who had finished growing completely.

Exclusion criteria: Subjects with a history of orthodontic treatment or orthognathic surgery, abnormalities in the number and morphology of teeth, and those who were medically compromised.

### 2.2. Cephalometric Analysis

All cephalometric tracings and measurements were performed by trained orthodontists in our department and double-checked by a certified orthodontist. The errors of the measurements were within 0.2 mm in our preliminary study. COA5 software (Rocky Mountain Morita, Tokyo, Japan) was used for cephalometric analysis. Seventeen landmarks were used for lateral cephalometric analysis, and 17 landmarks were used (Table 1). Nine fundamental planes were measured: the SN plane (SN), Frankfort horizontal plane (FH), palatal plane (PP), occlusal plane (OP, defined as a plane drawn between the cusp of the first molars and central incisors), mandibular plane (MP, defined as a tangent to the lower border of the mandible through the Me), ramus plane (RP, defined as a tangent to the posterior border of the ramus through Ar), Ar-A plane (Ar-A), Lr-B plane (Lr-B), and Rickett’s E-line (defined as the line between the soft tissue chin and the tip of the nose) (Figure 1).

Fourteen items of skeletal pattern (5 liner and 9 angler), 16 items of denture pattern (6 liner and 10 angler), and 3 items of soft tissue (2 liner and 1 angler) were used as measurement items (Table 1).

Furthermore, frontal cephalometric analysis [23] was performed using these four landmarks (Table 2 and Figure 2). The facial midline was drawn, and mandibular deviation was evaluated. In other words, a line perpendicular to the line connecting the left and right Lo points through the Nc was defined as the midline of the face, and the mandibular deviation was evaluated as the distance from the Me to the midline of ±2 mm or more (Figure 2).

### 2.3. Statistical Analysis

A prior sample size was calculated on the basis of the findings of a previous study [24] for cephalometric analysis of skeletal open-bite patients using the *t*-test. The analysis was performed using G*Power 3.1 software (Heinrich Heine University, Dusseldorf, Germany) with the power of the statistical test, and the error probability was set at 80% and 0.05. According to the results, a sample size of 36 subjects (18 ICR and 18 control groups) was considered appropriate.

Normality was analyzed and confirmed using the Shapiro–Wilk normality test. After the F-test of equality of variances, Student’s *t*-test was performed to examine the differences in cephalometric measurements between the ICR and control groups. Analysis was performed using Microsoft Excel (Microsoft Corp., Redmond, WA, USA). The level of significance was set at 0.05 for all analyses.

## 3. Results

### 3.1. Clinical Findings on Malocclusion

In all ICR patients, the molar relationship was bilateral Angle class II. Furthermore, mandibular deviation to the right side was observed in five cases and to the left side in four cases, and mandibular deviation with respect to the facial midline was confirmed by frontal cephalometric analysis (Table 3).

### 3.2. TMJ Findings

As for TMJ symptoms, crepitus of the left TMJ was observed in one case, and bilateral click and opening pain were observed in another case. All the other patients were asymptomatic. TMJ-MRI revealed ADDWOR on the right side and anterior disk displacement with reduction (ADDWR) on the left side in two cases. Bilateral ADDWOR was observed in another 16 cases. Morphological abnormalities of the mandibular condyle were observed on both sides in 17 cases, the left side in one case. All TMJs showed deformation and atrophy of the mandibular condyle, and bone marrow signal alterations on MRI, showing typical ICR findings (Table 3).

### 3.3. Evaluation of Skeletal Pattern

There was no significant difference in the length of the S-N, which is the horizontal reference plane of the skull, between the ICR and control groups (*p* > 0.05). In addition, the anterior facial height was evaluated using N-A and A-B. Although N-A, an index of upper facial height, did not differ significantly between the ICR and control groups (*p* > 0.05), A-B, an index of lower facial height, in the ICR group was significantly greater than that in the control group (*p* < 0.05). The length of the mandible and Ar-Lr and the length of the posterior margin of the mandibular branch in the ICR group were significantly smaller than those in the control group (Go-Me, Ar-Lr, *p* < 0.05) (Table 4).

In the evaluation of the horizontal relationship between the upper and lower jaws, the ANB angle, SNP angle, facial angle, and angle of convexity in the ICR group showed a significantly higher tendency for skeletal maxillary protrusion than those in the control group (*p* < 0.05). Since all SNA, SNB, and SNP angles were significantly smaller than those in the control group (*p* < 0.05), both the maxilla and mandible were posterior to the skull base in the ICR group (Table 4).

In the evaluation of the vertical relationship between the upper and lower jaws, the FMA, SN/MP, and SN to the ramus plane, and gonial angle in the ICR group were significantly greater than those in the control group (*p* < 0.05), and clockwise rotation of the mandible was observed in the ICR group (Table 4).

### 3.4. Evaluation of Denture Pattern

The overjet was significantly larger (*p* < 0.05) and the overbite was significantly smaller (*p* < 0.05) in the ICR group than in the control group. There were no significant differences between the ICR and control groups in the vertical distances from the upper and lower central incisal edges to the Ar-A, maxillary reference plane, or mandibular reference plane (*p* > 0.05). In addition, there were no significant differences between the ICR and control groups in the vertical distance from the upper and lower first molars to Ar-A and Lr-B, respectively (*p* > 0.05). SN to OP and PP to OP were used to evaluate the inclination of the occlusal plane with respect to the reference plane. The occlusal plane was significantly clockwise with respect to the skull base and palatal plane in the ICR group compared to that in the control group (*p* < 0.05) (Table 5).

In the evaluation of the inclination of the upper central incisors, no significant differences were observed between the ICR and control groups in U1 to SN, U1 to FH, U1 to PP, or U1 to OP (*p* > 0.05). That is, the inclination of the maxillary central incisors in the ICR group was within the normal range with respect to the skull base, FH, palatal, and occlusal planes. L1 to FH, L1 to OP, and L1 to MP were used to evaluate inclination of the lower central incisors. The lower central incisors in the ICR group were significantly proclined with respect to the FH and palatal planes compared to those in the control group (*p* < 0.05). There was no significant difference in L1 to MP between the ICR and control groups (*p* > 0.05). The interincisal angle in the ICR group was significantly smaller than that in the control group (*p* < 0.05) (Table 5).

### 3.5. Evaluation of Soft Tissue

The vertical distances of the upper and lower lips to the E-line in the ICR group were significantly greater than those in the control group (*p* < 0.05); that is, protrusions of the upper and lower lips were observed in the ICR patients. In addition, the Z-angle was significantly smaller in the ICR group than in the control group (*p* < 0.05); that is, protrusion of the lower lip and retruded chin were observed in ICR patients (Table 6).

## 4. Discussion

In the present study, all subjects in the ICR group had Angle class II malocclusion, and nine (50%) had an overjet of 7 mm or more. This is largely consistent with the results of previous studies [10].

In the evaluation of the skeletal pattern, there were no significant differences between the ICR and control groups in the depth and height from the skull base to the maxilla. Compared with the control group, ICR patients exhibited skeletal maxillary protrusion in the horizontal relationship between the maxilla and the mandible. In addition, both the maxilla and mandible were significantly posterior to the skull base in the ICR group than in the control group, suggesting that skeletal maxillary protrusion in ICR patients was mainly due to retrusion of the mandible.

Nine subjects (50%) in the ICR group had an anterior open bite (negative overbite). This result is in good agreement with those of previous studies [10,25]. A vertical assessment of the maxillofacial skeleton suggested that N-A, the upper facial height, in the ICR group was not significantly different from that in the control group, whereas in the ICR group, the lower facial height was significantly longer in the ICR group than in the control group. As the size of the mandible in the ICR group was significantly smaller than that in the control group in terms of both mandibular body length and ramus height, excessive lower facial height was not due to downward overgrowth of the mandible. Since an excessive gonial angle and significant clockwise rotation of the mandibular and ramus planes were observed in ICR patients, the clockwise rotation of the mandible possibly caused point B to move downward, resulting in an excessively lower facial height.

Mandibular condylar resorption in ICR causes a decrease in mandibular condyle volume and mandibular condylar height, resulting in mandibular retrusion independent of TMJ pain and noise [25]. It has been suggested that maxillofacial morphology, such as skeletal open bite with steep mandibular and occlusal planes and/or skeletal maxillary protrusion with increased overjet and decreased SNB angle, can be induced [9,25]. Therefore, if both mandibular condyles are evenly affected, backward and downward rotation of the mandible results in malocclusion with skeletal class II, high angle, and open bite [16]. However, if uneven resorption of the mandibular condyles occurs even with unilateral or bilateral ICR, mandibular deviation and discrepancy of the midline between the upper and lower dentitions can occur, leading to crossbite or unilateral skeletal open bite [16].

In the present study, unilateral ICR was observed in one case in which the TMJ on the opposite side was normal and the mandible deviated toward the affected side. In contrast, of the 17 cases of bilateral ICR, 5 cases had right-sided mandibular deviation and 3 cases had left-sided mandibular deviation. Therefore, in such cases, it is speculated that the degree of resorption in the mandibular condyle may have been uneven between the right and left sides.

In the evaluation of the dental pattern, a significantly greater overjet and smaller overbite were observed in the ICR group in comparison with the control group. In contrast, the inclination of the central incisor was normal in the ICR group, and there was no tendency for alveolar maxillary protrusion. Therefore, the excessive overjet in the ICR group was considered to be due to the retruded position of the lower central incisors rather than the protruded position of the upper central incisors. In addition, the vertical distances of the upper and lower central incisors to the reference planes in the ICR group were within the normal range, indicating that the small overbite was not due to the intrusion of the incisors. The inclination of the lower central incisor with respect to the FH and occlusal planes in the ICR group was significantly higher than that in the control group; however, the inclination with respect to the mandibular plane was not significantly different between the two groups. From this result, it is considered that the labial inclination of the lower central incisor was normal for the mandible and was caused by clockwise rotation of the mandible. Therefore, the ICR group showed a significantly smaller interincisor angle than the control group, possibly because the lower central incisor axis proclined with respect to the upper central incisor axis due to the clockwise rotation of the mandible.

In general, in malocclusion with skeletal discrepancy, dentoalveolar compensation, a system that attempts to maintain normal interarch relationships under varying jaw relationships, is often recognized [26]. In an open skeletal bite, the upper and lower anterior teeth are extruded as dentoalveolar compensation [27]. In the present study, no protrusion of the upper and lower central incisors was observed in the ICR group compared with the control group. ICR is characterized by continuous and rapid resorption of the mandibular condyle, which is speculated to reduce the likelihood of dental compensation less likely to occur. Furthermore, in an open skeletal bite, the upper and lower molars are extruded according to the clockwise rotation of the mandible [27]. The vertical dentoalveolar compensation in adult skeletal open bite patients involves extrusion of the upper and lower incisors, with the mandibular incisors playing a more important role [27,28]. In the present study, there was no significant difference in the vertical distance between the upper and lower first molars of the upper and lower molars to the reference plane in the ICR group compared with the control group. Mandibular rotation was observed in the ICR group because the mandibular condyle was abnormally absorbed, resulting in shortened mandibular ramus height (Ar-Lr). Therefore, it is considered that the mandible did not simply rotate clockwise around the mandibular condyle as in a normal skeletal open bite but rotated further posteriorly due to the shortening of the mandibular ramus. However, no previous study has investigated dentoalveolar competition in ICR patients with skeletal open bites has been available. Therefore, further studies are warranted.

According to the clockwise rotation and retruded position of the mandible, as shown in the cephalometric analysis of the skeletal pattern, significant effects on the profile were also observed. Soft tissue evaluation revealed significant upper and lower lip protrusions and chin retrusion in the ICR mice.

The male-to-female ratio of ICR patients is 1:9 [9] or 1:16 [10], which occurs more often in women. In particular, it is often observed during adolescent growth [9,11], but it is suggested that absorption is rarely sustained after the age of 40 [13]. Therefore, the involvement of sex hormones in ICR onset is strongly suspected. In a survey of 88 TMD specialists using a questionnaire [10], 42 of 94 female patients diagnosed with ICR were taking contraceptives. It has been suggested that in the TMJ, symptomatic patients have higher levels of estrogen and progesterone receptors in the articular disc than asymptomatic patients [29]. Therefore, the effects of estrogen and progesterone on bone metabolism have been suggested [19,29]. In this study, all subjects in the ICR group were women, which is consistent with the results of previous studies. However, no history of contraceptive medication use or sex hormone imbalance was confirmed in any of the ICR patients. A limitation of this study is that the number of subjects was not sufficient for further study. Owing to the extremely small number of ICR patients, it is necessary to consider multi-institutional joint research in the future.

From the results of this study, it is necessary to consider the possibility of abnormal mandibular condylar resorption in the diagnosis of severe skeletal open bite and maxillary protrusion accompanied by mandibular retrusion and to confirm the presence or absence of a continuous progression of skeletal changes such as mandibular deviation and clockwise rotation. Most previous reports on maxillofacial morphology in ICR patients have targeted post-growth subjects, and thus the characteristics of maxillofacial morphology in younger patients with ICR remain unknown [10,12,17,19]. Therefore, it is considered effective to confirm the history of TMJ symptoms and perform appropriate imaging examination of the TMJ as necessary for the early detection of ICR.

## 5. Conclusions

All subjects in the ICR group had Angle class II malocclusion and half had an overjet of 7 mm or more. The ICR group had a significantly stronger tendency toward skeletal class II than the control group, mainly due to retrusion of the mandible. In the ICR group, there was a tendency for skeletal open bite due to significantly greater clockwise rotation of the mandible than in the control group. There was no significant difference between the two groups in the inclination of the upper and lower central incisors and extrusion of the upper and lower central incisors and first molars, that is, no dentoalveolar compensation for skeletal discrepancy was observed in the ICR group. Further research is necessary to elucidate the pathological conditions and to establish a treatment method for ICR.

## Figures and Tables

**Figure 1 jcm-11-00952-f001:**
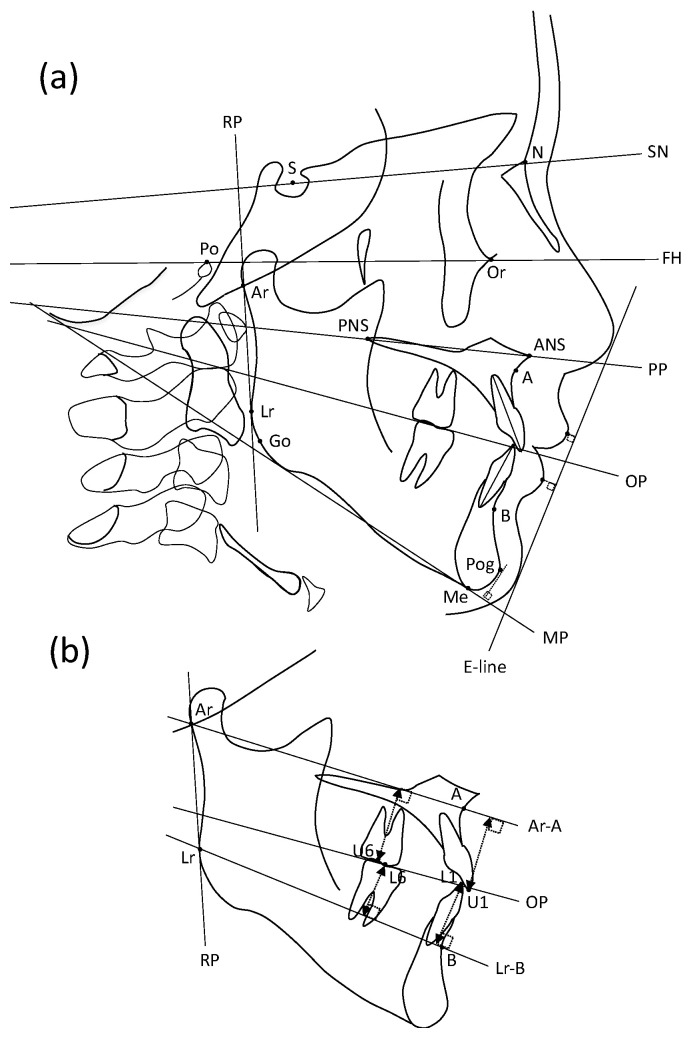
Landmarks and fundamental planes for the lateral cephalometric analysis. Seventeen landmarks and nine fundamental planes were used in the analysis. (**a**) Landmarks for all maxillofacial measurement items. (**b**) Landmarks for evaluation of the vertical position of incisors and molars.

**Figure 2 jcm-11-00952-f002:**
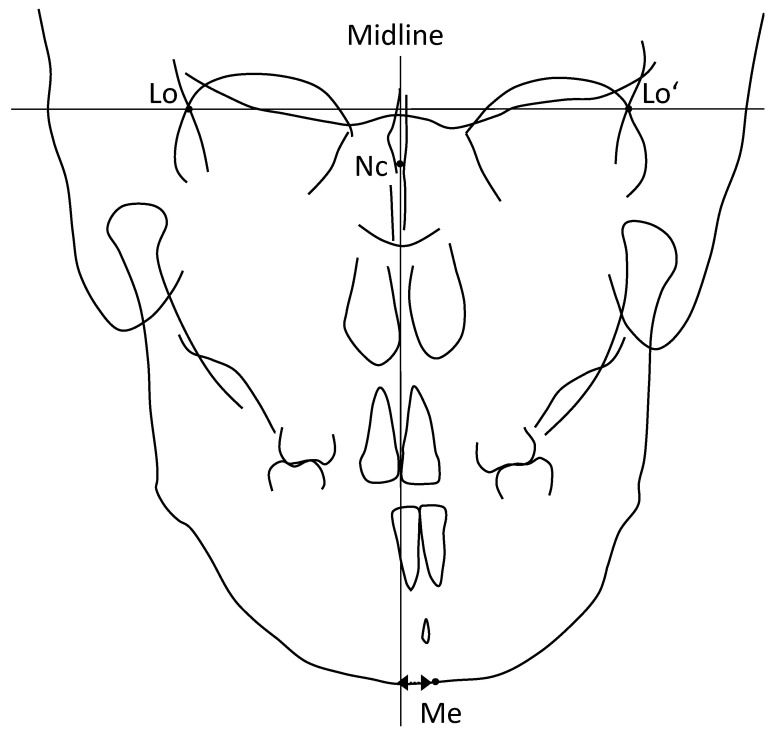
Landmarks and fundamental planes for the frontal cephalometric analysis. Four landmarks and two fundamental lines were used for the analysis.

**Table 1 jcm-11-00952-t001:** Cephalometric measurements.

Measurement	Definition
Skeletal pattern: linear	
S-N	Distance between S and N
N-A	Distance between N and A
A-B	Distance between A and B
Go-Me	Distance between Go and Me
Ar-Lr	Distance between Ar and Lr
Skeletal pattern: angular	
Facial angle	Angle between FH plane and N-Pog line
Angel of convexity	Angle between NA line and A-Pog line
SN to MP	Angle between SN plane and MP
FMA	Angle between FH plane and MP
Gonial angle	Angle between Ramus plane and MP
SNP	Angle between SN plane and N-Pog
SNA	Angle between SN plane and NA line
SNB	Angle between SN plane and NB line
ANB	Difference between SNA and SNB
Denture pattern: linear	
Overjet	Distance between the incisal edges of the upper and lower central incisors measured at the level of OP
Overbite	Distance between the incisal edges of the upper and lower central incisors, measured perpendicular to OP
U1 to Ar-A	Ar-A line to U1
L1 to Lr-B	Lr-B line to L1
U6 to Ar-A	Ar-A line to U6
L6 to Lr-B	Lr-B line to L6
Denture pattern: angular	
SN to OP	Angle between SN plane and OP
PP to OP	Angle between PP and OP
U1 to SN	Angle between long axis of U1 and SN plane
U1 to FH	Angle between long axis of U1 and FH plane
U1 to PP	Angle between long axis of U1 and PP
U1 to OP	Angle between long axis of U1 and OP
L1 to FH (FMIA)	Angle between long axis of L1 and FH plane
L1 to OP	Angle between long axis of L1 and OP
L1 to MP (IMPA)	Angle between long axis of L1 and MP
Interincisal angle	Angle between long axis of U1 and L1
Soft tissue: linear	
E-line to ULP	Retruded or protruded upper lip relative to E-line
E-line to LLP	Retruded or protruded lower lip relative to E-line
Soft tissue: angular	
Z-angle	Angle formed by the intersection of FH plane and a line connecting soft tissue pogonion and the most protrusive lip point

**Table 2 jcm-11-00952-t002:** Cephalometric landmarks.

Landmark	Abbreviation	Definition
Lateral cephalometric analysis		
Sella	S	Mid-point of sella turcica
Nasion	N	Most anterior point of frontonasal suture
Porion	Po	Most superior point of the external auditory meatus
Orbitale	Or	Most inferior point on the lower border of the bony orbit
Ariculare	Ar	Intersecting point of the inferior cranial base with mandibular posterior border
Posterior nasal spine	PNS	Most posterior point of hard palate
Anterior nasal spine	ANS	Anterior point of maxilla
A-point	A	Deepest anterior point on maxilla anterior concavity
B-point	B	Deepest anterior point on mandibular symphysis
Pogonion		Most anterior point on mandibular symphysis
Lr	Lr	A contact point of ramus plane to mandibular ramus
Gonion	Go	Intersection of the lines tangent to the posterior margin of the ascending ramus and the MP
Menton	Me	Most inferior point on mandibular symphysis
Upper incisor	U1	Edge of the upper central incisor
Lower incisor	L1	Edge of the lower central incisor
Upper first molar	U6	Mesio-distal mid-point of the upper first molar outline
Lower first molar	L6	Mesio-distal mid-point of the lower first molar outline
Frontal cephalometric analysis		
Latero-orbitale	Lo	Intersecting point between the external orbital contour laterally and the oblique line
Crista galli	Nc	The most narrowed part of the neck of crista galli
Menton	Me	The point on the inferior border of bony chin right below the genial tubercle

**Table 3 jcm-11-00952-t003:** Clinical and magnetic resonance imaging findings of ICR patients.

Subject	Age	Sex	Overjet (mm)	Overbite (mm)	Molar Relationship		TMJ Symptoms		Disk Displacement		Condylar Deformity		Mandibular Deviation
					R	L	R	L	R	L	R	L	
1	23Y6M	F	5.4	−1.2	II	II	N	N	ADDWOR	ADDWOR	Y	Y	N
2	31Y5M	F	8.9	−4.4	II	II	N	N	ADDWOR	ADDWOR	Y	Y	N
3	23Y3M	F	11.7	−5.2	II	II	N	N	ADDWOR	ADDWOR	Y	Y	L
4	26Y7M	F	16.5	0.6	II	II	N	N	ADDWOR	ADDWOR	Y	Y	N
5	29Y8M	F	6.1	−6.6	II	II	N	N	ADDWOR	ADDWR	Y	Y	N
6	20Y3M	F	4.1	1.7	II	II	N	N	ADDWOR	ADDWR	Y	Y	R
7	34Y2M	F	3.3	1.3	II	II	N	N	ADDWOR	ADDWOR	Y	Y	R
8	21Y8M	F	7.2	0.8	II	II	N	N	ADDWOR	ADDWOR	Y	Y	N
9	28Y6M	F	6.6	−1.6	II	II	N	N	ADDWOR	ADDWOR	Y	Y	L
10	26Y6M	F	5.2	−2.5	II	II	N	N	ADDWOR	ADDWOR	Y	Y	N
11	27Y9M	F	3.8	2.2	II	II	N	Creptus	N	ADDWOR	N	Y	L
12	16Y6M	F	9	4.8	II	II	Click, pain	Click, pain	ADDWOR	ADDWOR	Y	Y	L
13	28Y2M	F	4.5	0.4	II	II	N	N	ADDWOR	ADDWOR	Y	Y	N
14	20Y6M	F	10.8	0.8	II	II	N	N	ADDWOR	ADDWOR	Y	Y	R
15	33Y11M	F	9.9	−2.1	II	II	N	N	ADDWOR	ADDWOR	Y	Y	R
16	17Y11M	F	12.5	3.8	II	II	N	N	ADDWOR	ADDWOR	Y	Y	N
17	20Y5M	F	6.5	−1.2	II	II	N	N	ADDWOR	ADDWOR	Y	Y	R
18	29Y0M	F	8.5	−0.5	II	II	N	N	ADDWOR	ADDWOR	Y	Y	N

**Table 4 jcm-11-00952-t004:** Skeletal pattern measurements of lateral cephalometric analysis in control and ICR groups.

Measurement	Control		ICR	
	Mean	SD	Mean	SD
Linear				
S-N	67.9	2.6	69.9	2.8
N-A	62.6	3.6	62.8	3.0
A-B	42.2	3.7	46.0 *	4.9
Go-Me	74.6	3.8	69.5 *	4.8
Ar-Lr	41.1	4.8	30.7 *	4.8
Angular				
Facial angle (FH to N-Pog)	85.0	4.1	80.1 *	5.3
Angel of convexity (N-A-Pog)	5.3	5.3	19.6 *	8.5
SN to MP	33.2	4.5	51.5 *	7.1
FMA (MP to FH)	28.8	6.0	41.6 *	6.8
Gonial angle	119.3	5.8	127.7 *	7.4
SN to Ramus plane	93.9	4.3	103.8 *	9.6
SNP	80.6	2.9	70.2 *	4.7
SNA	83.1	2.5	80.1 *	3.8
SNB	80.2	2.6	71.3 *	4.2
ANB	2.9	2.2	8.9 *	3.2

* *p* < 0.05.

**Table 5 jcm-11-00952-t005:** Denture pattern measurements of lateral cephalometric analysis in control and ICR groups.

Measurement	Control		ICR	
	Mean	SD	Mean	SD
Linear				
Overjet	3.5	0.8	7.8 *	3.5
Overbite	2.0	1.1	−0.5 *	3
U1 to Ar-A	23.0	2.1	23.7	3.7
L1 to Lr-B	20.5	2.6	22.6	3.3
U6 to Ar-A	24.4	2.1	24.2	3.1
L6 to Lr-B	16.1	2.3	16.0	2.3
Angular				
SN to OP	17.6	3.3	25.1 *	5.1
PP to OP	7.7	2.5	14.1 *	4.1
U1 to SN	103.5	11.1	105.7	5.8
U1 to FH	110.0	7.0	113.4	11.2
U1 to PP	115.6	5.0	114.4	10.8
U1 to OP	56.7	4.6	51.4	8.4
L1 to FH (FMIA)	56.6	6.9	43.6 *	10.5
L1 to OP	69.9	6.7	58.7 *	8.6
L1 to MP (IMPA)	94.5	7.7	94.8	8.9
Interincisal angle	126.6	10.3	110.1 *	13.8

* *p* < 0.05.

**Table 6 jcm-11-00952-t006:** Soft tissue measurements of lateral cephalometric analysis in control and ICR groups.

Measurement	Control		ICR	
	Mean	SD	Mean	SD
Linear				
E-line to ULP	−1.7	2.3	4.5 *	4.7
E-line to LLP	−0.3	2.5	5.3 *	4.8
Angular				
Z-angle	71.7	7.8	54.0 *	14.5

* *p* < 0.05.

## Data Availability

Due to the nature of this research, participants of this study did not agree for their data to be shared publicly, so supporting data is not available.

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
