# Peer review of "Characteristics of the Maxillofacial Morphology in Patients with Idiopathic Mandibular Condylar Resorption"

_jcm, 2022, doi:10.3390/jcm11040952_

Round 1
Reviewer 1 Report
The article that studies the characteristics of the maxillofacial morphology in patients with idiopathic mandibular condylar resorption is not fully original but I think adds something interesting to the present literature on the topic.
However I found quite difficult to go through the article, the structure of the article is acceptable but lots of sentences are not clear, probably in part due to the English style and accuracy.
My comments:
INTRODUCTION
Several sections are not clear (lines: 31-33,45-52 ,66-74), particularly is not clear why you used adult patients and not growing patients, you should check and re-write the introduction where is needed.
MATERIALS AND METHOD
Samples
-Inclusion and exclusion criteria are not clear both for study and control group. -How do you select the control group? Did you match it for age and sex with the sample group? You should explain it in the manuscript.
Method
You should describe who traced and analyzed the lateral and PA ceph, if he /she was a student or an orthodontist, you should explain if there was only 1 operator or more. Did you evaluate the intra-rater (and inter) reliability? These should be added in order to discuss the bias related with the method.
Statistical analysis
-The sample size is not big (even if we have to consider that ICR is rare), how do you decide to use the T-test to compare the group. Did you study the distribution of the sample? It should be discussed it in the manuscript text.
RESULTS
The mean age and the percentage of males/females are not well described both for study and control group. You should describe it in the manuscript text or create a table as you did for the study group.
DISCUSSION
I think it is a bit too long and it is not focus enough on the important aspects:
- I think that section (306-323) discussing about the causes of ICR could be shorten
- I think that it should be discussed more about the fact that there is not dental compensation in ICR patients, you wrote :
“ICR is characterized by continuous and rapid resorption of the mandibular condyle, which is speculated to be a factor that reduces the likelihood of dental compensation less likely to occur. “
You should write more about this and add references to support it.
- You also wrote that
“Especially in young patients, the effect of TMJ pathology on maxillofacial morphology is often not significant,….”
Could you please explain this a little better and add references that support it.
Author Response
My comments: Thank you very much for your valuable comment. The article has been revised according to the comments, and the text has been re-checked by a native English speaker.
INTRODUCTION
(Reviewer’s comment) Several sections are not clear (lines: 31-33,45-52 ,66-74), particularly it is not clear why you used adult patients and not growing patients. You should check and re-write the introduction where is needed.
> I appreciate the reviewer’s suggestion. The text has been revised as follows:
Lines 31-39, “Among the pathological conditions with mandibular resorption, ICR is diagnosed when the possibility of all other diseases is excluded [3]. Pathological conditions related to mandibular condylar resorption include systemic diseases such as collagen disease, dermatomyositis, systemic sclerosis, growth hormone deficiency, and endocrine disorders, as well as local lesions such as trauma and a history of TMJ surgery [4-6]. In addition, ICR is clearly distinguished from mandibular condyle dysplasia, in which the growth and development of the mandibular condyle is inhibited for some reason [7]. ICR is localized to the TMJ and is not accompanied by arthritis or synovitis; therefore, it is also clearly distinguished from juvenile arthritis [8].”
Lines 42-67, “It has been reported that the progression of ICR may cause occlusion and skeletal muscle instability, leading to maxillofacial deformity and TMJ dysfunction and pain [9, 10]. However, TMJ symptoms in ICR are slight in many cases [11], and 25% are asymptomatic [12]. A possible reason for this could be that in ICR mice, enlarged joint space due to hyperplasia of synovial tissue makes the displaced articular disk movement easier during jaw movement [12]. As another study suggested no abnormalities in the synovial tissue of ICR joints, the details remains unclear [13].”
Lines 82-101, “However, the causal relationship between ICR and malocclusion requires further examination. This is because the exact time of onset of the pathological condition is unknown, making it difficult to accurately assess the effect of ICR on maxillofacial growth in most patients. In our department, the onset of ICR has been observed since the early teens; however, the onset of ICR in teens has not been confirmed. Since the number of growing patients in our department is small [22], it is possible that the initial pathological status of ICR has not been completely detected. In addition, since there are individual differences in maxillofacial skeletal growth during the growth period, it is difficult to have an appropriate control group for the young ICR patient group. On the other hand, in adult ICR patients, as the pathologic status progresses, aggravation of skeletal open bite accompanied by mandibular retrusion is often observed, and the pathological condition is relatively easy to detect. The control group was relatively easy to set up. Therefore, in the present study, we examined the characteristics of maxillofacial morphology in ICR mice after the completion of growth.
We hypothesized that the maxillofacial morphology of ICR patients differs from that of individuals with normal occlusion without temporomandibular disorders (TMDs). Since all the patients diagnosed with ICR in our department were female [22], patients in their late teens or later whose growth was confirmed to have completed were targeted. The purpose of this study was to compare maxillofacial morphology between ICR patients and control subjects of the same age and sex who have normal occlusion without TMDs.”
MATERIALS AND METHOD
Samples
(Reviewer’s comment) -Inclusion and exclusion criteria are not clear both for study and control group. -How do you select the control group? Did you match it for age and sex with the sample group? You should explain it in the manuscript.
> Thank you for your valuable comment. The text has been revised to include additional information as follows:
Lines 151-169, “Of the 309 patients, 268 (3.8% of all orthodontic patients and 86.8% of patients with mandibular condylar deformity) were diagnosed with TMJ-OA, whereas 25 were diagnosed with ICR (0.36% of all orthodontic patients and 8.1% of patients with mandibular condylar deformity). The diagnosis of ICR was made when continuous absorption on both sides or one of the mandibular condyles was shown on the TMJ-MRI, and the possibility of other diseases was excluded. Of these, 24 were diagnosed with ICR by TMJ-MRI at the first examination, and growing patients were excluded, and 18 (18 female and 0 male, age, mean ± SD, 25.5 ± 5.3 years) were selected as ICR group subjects in this study.
The control group consisted of 18 female volunteers in 4th grade (ages, mean ± SD, 24.1 ± 1.2 years) from dental students in Hiroshima University School of Dentistry from 2016 to 2019 (total 212 students) who consented to participate in the study and were judged to have normal occlusion and no TMDs. Lateral and frontal cephalograms obtained under the same conditions were used for both the ICR and control groups.
Inclusion criteria: Those who met the above conditions and who had finished growing completely.
Exclusion criteria: Subjects with a history of orthodontic treatment or orthognathic surgery, abnormalities in the number and morphology of teeth, and those who were medically compromised.
Method
(Reviewer’s comment) You should describe who traced and analyzed the lateral and PA ceph, if he /she was a student or an orthodontist, you should explain if there was only 1 operator or more. Did you evaluate the intra-rater (and inter) reliability? These should be added in order to discuss the bias related with the method.
> Thank you for pointing this out. The text has been revised to include additional information as follows:
Lines 173-176, “All cephalometric tracings and measurements were performed by trained orthodontists in our department and double-checked by a certified orthodontist. The measurement errors were within 0.2 mm in our preliminary study.”
Statistical analysis
(Reviewer’s comment) -The sample size is not big (even if we have to consider that ICR is rare), how do you decide to use the T-test to compare the group. Did you study the distribution of the sample? It should be discussed it in the manuscript text.
> We appreciate the reviewer’s valuable comments. The text has been revised to include the additional reference and Tables 4, 5, and 6 were revised with the error probability set at 0.05.
Lines 211-224,
“2.3. Statistical analysis
A prior sample size was calculated based on a previous study [24] for cephalometric analysis of skeletal open-bite patients using a t-test. The analysis was performed using G*Power 3.1 software (Heinrich Heine University, Dusseldorf, Germany) with the power of the statistical test, and the error probability was set at 80% and 0.05, respectively. According to the results, a sample size of 36 subjects (18 ICR and 18 control groups) was considered appropriate.
Normality was analyzed and confirmed using the Shapiro-Wilk normality test. After the F-test of equality of variances, Student’s t-test was performed to examine the differences in cephalometric measurements between the ICR and control groups. Analysis was performed using Microsoft Excel (Microsoft Corp., Redmond, WA, USA). The level of significance was set at P < 0.05.”
(Additional references)
- Akl, H.E., Abouelezz, A.M., El Sharaby, F.A., El-Beialy, A.R., El-Ghafour, M.A. Force magnitude as a variable in maxillary buccal segment intrusion in adult patients with skeletal open bite. Angle Orthod. 2020, 90(4), 507-515.
RESULTS
(Reviewer’s comment) The mean age and the percentage of males/females are not well described both for study and control group. You should describe it in the manuscript text or create a table as you did for the study group.
> Thank you very much for your comment. The Materials and Methods section has been revised as shown above.
DISCUSSION
(Reviewer’s comment) I think it is a bit too long and it is not focus enough on the important aspects:
- I think that section (306-323) discussing the causes of ICR could be shortened.
> Thank you for your valuable advice. The following text and references have been removed:
Lines 306-323, “Regarding the cause of ICR, abnormal mandibular condylar resorption after orthognathic surgery is well known, but the reported incidence varies from 1.2% to 20.2% [20]. According to the results of 32 articles published between 1970 and 2014, 67.8% of orthognathic surgeries that developed mandibular condylar resorption were two-jaw surgeries [20]. Although the onset of ICR has been reported both after mandibular advancement [21, 27, 28] and after two-jaw surgery [29-31], the actual onset mechanism remains unknown.”
(Reviewer’s comment)
- I think that it should be discussed more about the fact that there is not dental compensation in ICR patients, you wrote:
“ICR is characterized by continuous and rapid resorption of the mandibular condyle, which is speculated to be a factor that reduces the likelihood of dental compensation less likely to occur. “
You should write more about this and add references to support it.
> We appreciate the reviewer’s valuable suggestion. The text was revised to include the additional reference as follows:
Lines 385-399, “ICR is characterized by continuous and rapid resorption of the mandibular condyle, which is speculated to be a factor that reduces the likelihood of dental compensation less likely to occur. Furthermore, in an open skeletal bite, the upper and lower molars are extruded according to the clockwise rotation of the mandible [27]. The vertical dentoalveolar compensation in adult skeletal open bite patients involves extrusion of the upper and lower incisors, with the mandibular incisors playing a more important role [27, 28]. In the present study, there was no significant difference in the vertical distance between the upper and lower first molars of the upper and lower molars to the reference plane in the ICR group compared with the control group. Mandibular rotation was observed in the ICR group because the mandibular condyle was abnormally absorbed, resulting in shortened mandibular ramus height (Ar-Lr). Therefore, it is considered that the mandible did not simply rotate clockwise around the mandibular condyle as in a normal skeletal open bite, but rotated further posteriorly due to the shortening of the mandibular ramus. However, no previous study has investigated dentoalveolar competition in ICR patients with skeletal open bites has been available. Therefore, further studies are needed.”
(Additional references)
- Kuitert, R.; Beckmann, S.; van Loenen, M.; Tuinzing, B.; Zentner, A. Dentoalveolar compensation in subjects with vertical skel-etal dysplasia. Am J Orthod Dentofacial Orthop. 2006, 129(5), 649-657.
(Reviewer’s comment)
- You also wrote that
“Especially in young patients, the effect of TMJ pathology on maxillofacial morphology is often not significant,….”
Could you please explain this a little better and add references that support it.
> We appreciate the reviewer’s valuable suggestion. The text has been revised as follows:
Lines 418-426, “From the results of this study, it is necessary to consider the possibility of abnormal mandibular condylar resorption in the diagnosis of severe skeletal open bite and maxillary protrusion accompanied by mandibular retrusion, and to confirm the presence or absence of a continuous progression of skeletal changes such as mandibular deviation and clockwise rotation. Most previous reports on maxillofacial morphology in ICR patients have targeted post-growth subjects, so the characteristics of maxillofacial morphology in younger patients with ICR remain unknown [10, 12, 17, 19]. Therefore, it is considered effective to confirm the history of TMJ symptoms and perform appropriate imaging examination of the TMJ as necessary for the early detection of ICR.”
Reviewer 2 Report
Introduction: add study hypothesis.
Material and methods: Was the sample calculation performed?
Results: Add the normality test plots of the study variables.
Discussion: What were the limitations of the study?
Author Response
My comments: Thank you very much for your valuable comment. The article has been revised according to the comments, and the text has been re-checked by a native English speaker.
(Reviewer’s comment) Introduction: add study hypothesis.
> I appreciate the reviewer’s suggestion and as such, the following text has been added;
Lines 96-97, “We hypothesized that the maxillofacial morphology of ICR patients differs from that of individuals with normal occlusion without temporomandibular disorders (TMDs).”
(Reviewer’s comment) Material and methods: Was the sample calculation performed?
> We appreciate the reviewer’s valuable comments. The text was revised with an additional reference as follows, and Tables 4, 5, and 6 were revised with the error probability set at 0.05.
Lines 212-217, “A prior sample size was calculated based on the findings of a previous study [24] regarding cephalometric analysis of skeletal open bite patients using t-test. The analysis was performed using G*Power 3.1 software (Heinrich Heine University, Dusseldorf, Germany) with the power of the statistical test, and the error probability was set at 80% and 0.05, respectively. According to the results, a sample size of 36 subjects (18 ICR and 18 control groups) was considered appropriate.”
(Reviewer’s comment) Results: Add the normality test plots of the study variables.
> Thank you for your valuable advice. The following text has been added:
Line 218, “The normality was analyzed and confirmed by Shapiro-Wilk normality test.”
Discussion: What were the limitations of the study?
> We appreciate the reviewer’s valuable suggestion. The following text has been added:
Lines 415-417, “A limitation of this study is that the number of subjects was not sufficient for further study. Owing to the extremely small number of ICR patients, it is necessary to consider multi-institutional joint research in the future.”

Round 2
Reviewer 1 Report
Thank you for the changes you have made to the manuscript. I can see the improvements from this version. I may recommend the journal to publish this manuscript after some revisions:
- The English style and accuracy are not sufficient yet .
- I did not find enough comments about the fact that there is not dental compensation in ICR patients.
Author Response
My comments: Thank you very much for valuable comments from reviewers. The article was revised according to the comments, and all texts were re-checked by a native English speaker.
(Reviewer’s comment)
What are the 4 landmarks? Table 2 has more than 4 landmarks so it is not clear exactly what this refers to?
> Thank you very much for the reviewer’s suggestion. Four landmarks mean Lo, Lo’, Nc and Me for frontal cephalogram ads shown in Figure 2. To make it clear, the texts was revised as follows;
“Furthermore, frontal cephalometric analysis [23] was performed using these four landmarks (Table 2 and Figure 2).”
(Reviewer’s comment)
This section is unclear as you start talking about 17 cases then 1 case in the same sentence. I will be more than happy to make further edits if you choose to revise this section further.
> We really appreciate the reviewer for the crucial indication. The texts were revised as follows;
“Morphological abnormalities of the mandibular condyle were observed on both sides in 17 cases, the left side in one case.”
(Reviewer’s comment)
I would not advise giving your Conclusions in bullet points, I would advise writing them in full sentences.
> Thank you very much for the reviewer’s suggestion. The texts were revised in full sentences.
“All subjects in the ICR group had Angle Class II malocclusion and half had an overjet of 7 mm or more. The ICR group had a significantly stronger tendency toward skeletal Class II than the control group, mainly due to retrusion of the mandible. In the ICR group, there was a tendency for skeletal open bite due to significantly greater clockwise rotation of the mandible than in the control group. There was no significant difference between the two groups in the inclination of the upper and lower central incisors and extrusion of the upper and lower central incisors and first molars; that is, no dentoalveolar compensation for skeletal discrepancy was observed in the ICR group. Further research is necessary to elucidate the pathological conditions and to establish a treatment method for ICR.”